# Assessment of correlates of hand hygiene compliance among final year medical students: a cross-sectional study in the Netherlands

Vicki Erasmus,[1] Suzie Otto [iD],[1] Emmely De Roos,[2] Rianne van Eijsden,[1] Margreet C Vos,[3] Alex Burdorf [iD],[1] Ed van Beeck[1]

[1] Department of Public Health, Erasmus MC, Rotterdam, The Netherlands
[2] Department of Internal Medicine, Erasmus MC, Rotterdam, The Netherlands
[3] Department of Medical Microbiology and Infectious Diseases, Erasmus MC Rotterdam, Rotterdam, Zuid-Holland, Netherlands

**Correspondence to**
Dr Vicki Erasmus;
v.erasmus@erasmusmc.nl

## ABSTRACT

**Objectives** To identify the factors that influence the hand hygiene compliance of final year medical students, using a theoretical behavioural framework.

**Design** Cross-sectional survey assessing self-reported compliance and its behavioural correlates.

**Setting** Internships of medical students in the Netherlands.

**Participants** 322 medical students of the Erasmus Medical Center were recruited over a period of 12 months during the Public Health internship, which is the final compulsory internship after an 18-month rotation schedule in all major specialities.

**Primary and secondary outcome measures** Behavioural factors influencing compliance to hand hygiene guidelines were measured by means of a questionnaire based on the Theory of Planned Behaviour and Social Ecological Models. Multiple linear regression analysis was used to identify the effect of including attitudes, social norms, self-efficacy, knowledge, risk perception and habit on hand hygiene compliance.

**Results** We included 313 students in the analysis (response rate 97%). The behavioural model explained 40% of the variance in self-reported compliance (adjusted $R^2$=0.40). Hand hygiene compliance was strongly influenced by attitudes (perceived outcomes of preventive actions), self-efficacy (perception of the ability to perform hand hygiene at the clinical ward) and habit, but was not associated with knowledge and risk perception.

**Conclusions** Targeting medical students' behaviour should focus on the empowerment of these juniors and provide them with evidence on the health benefits of prevention, rather than increasing their factual knowledge of procedures. Clinical teaching environments could help them form good patient safety habits during this vital phase of their career.

## Strengths and limitations of this study

► This study uses a hand hygiene questionnaire, based on insights from both the Theory of Planned Behaviour and Social Ecological Models.
► The hand hygiene behaviour of medical students is investigated from a behavioural perspective in a large sample of students at the same stage of their internships.
► The main limitations of this study are its cross-sectional design and the self-reported compliance with hand hygiene guidelines.

## INTRODUCTION

Patient safety made its entrance into the fields of medical research and practice in the last decade of the twentieth century.[1–3] Ever since, patient safety has become a rising priority among health institutions, governments and insurance companies, who are all seeking to reduce the human and financial costs of preventable adverse events.[4] Prioritising these events based on their impact and frequency of occurrence, shows that — next to surgical[5–7] and medication procedures[8 9] — infection prevention is a key element to improve patient safety.[10 11] In high-income countries, 3.5% to 12% of all hospitalised patients contract one or more healthcare-associated infections (HAI), while approximately 20% to 30% of patients in critical care are affected.[12] This high incidence of HAI not only accounts for prolonged hospital stay and preventable morbidity and mortality of patients and additional costs, but also enlarges the global threat to human health due to the emergence of multiresistant bacteria by using antibiotics needed to treat these infections.[13 14]

Adequate hand hygiene by all medical professionals has been recognised as an eminent measure to reduce transmission of (multiresistant) pathogens.[14] However, adherence to hand hygiene guidelines, as with many quality improvement guidelines, is low, in particular among physicians[15] and medical students.[16] Medical professionals' patient safety practices, including their (non)adherence to hand hygiene guidelines, should be traced back to how future

medical professionals are trained today.[17–20] The inclusion of medical students in patient safety initiatives is vital, because students shape their behaviour and form habits during their internships, making them a priority group to be targeted in interventions.[21]

In the last decades, research into the behavioural factors influencing the hand hygiene behaviour of healthcare professionals has received growing attention.[22 23] Assessment of hand hygiene behavioural factors has often been guided by behavioural theories, in particular the Theory of Planned Behaviour (TPB).[24–28] The TPB states that individual behaviour is influenced by attitudes, social norms (ie, what people around you think and do) and self-efficacy (ie, whether you feel that you are able to perform the behaviour).[29] This theory has been used to explain the behaviour of physicians and nurses, but not yet that of medical students. Only a few studies into factors explaining hand hygiene compliance among medical students have been conducted so far, most lacking a theoretical behavioural framework or restricted to environmental factors such as access to facilities (low access associated with low compliance) and compliance of peers and superiors (low compliance of medical staff associated with low student compliance).[30 31] Van de Mortel et al[16 32 33] investigated factors associated with the hand hygiene behavioural of students in Australia, Italy and Greece, showing that knowledge was regularly insufficient, differences in hand hygiene attitudes and behaviours between future doctors and nurses are already present at an undergraduate level and that better hand hygiene education is necessary. Therefore, to date knowledge is still rather limited on the behavioural factors that influence patient safety behaviours, including hand hygiene, of our next generation of physicians.[34] These insights can help clinical educators to promote patient safety behaviour among medical students and thereby improve patient safety in the near future.

In this study we sought to identify correlates of the hand hygiene compliance of sixth-year medical students, using a theoretical behavioural framework.

## METHODS
### Setting and participants
Over a period of 12 months we recruited a class of 322 medical interns; a researcher visited the class room during regular lessons and invited students to complete the paper and pencil questionnaire. All students were enrolled in their last compulsory internship (Public Health) at the Erasmus Medical Center (MC), Rotterdam, (a 1320 bed university hospital). Every 2 weeks, a group of 10 to 15 students started their Public Health training and were invited to complete a questionnaire. Before this internship, the students followed an 18-month rotation schedule in all major specialities: internal medicine, surgery, paediatrics, gynaecology, neurology, psychiatry, ENT (Ear, Nose and Throat medicine), dermatology and general practice. The interns rotated among the

university medical centre and 20 non-academic hospitals within the region South-West Netherlands. These institutions serve over 6.3 million people of various social and ethnic backgrounds in a mixed urban and rural area. In the second year of the study, undergraduate medical students completed the compulsory 2-hour practical module 'Basic Hygiene' of the Unit Infection Prevention (Erasmus MC), in which they are taught the principles of hand hygiene, among other things.

A researcher or research assistant visited the class room during the lesson and invited all students to participate. Students only received a questionnaire if they indicated they were willing to participate. Questionnaires were completed anonymously and the researcher returned later to collect the completed questionnaires. This study is part of the hand hygiene project, which was provided a waiver by Institutional Review Board of Erasmus MC Rotterdam.

### Patient and public involvement
Patients and the public were not involved in the design of the study, or in the recruitment to and conduct of the study.

## BEHAVIOURAL THEORY AND QUESTIONNAIRE
We developed the questionnaire used in this study for a larger national study on the determinants of hand hygiene compliance. This questionnaire is based on a translated version of the Hand hygiene Assessment Instrument (HAI)[35] with additional constructs identified by qualitative research among physicians, nurses and medical students.[17] In particular, adequate knowledge of hand hygiene guidelines, hand hygiene as a habitual behaviour and risk perception (risk of contracting an infection yourself, and risk of patient contracting an infection) were identified in a qualitative study we performed. The HAI was translated into Dutch by two Dutch speakers and translated back into English by a native speaker to ensure the content had not changed. Experts in the field of behavioural science (n=3) and infection control (n=3) examined all items to ensure face and content validity.

The questionnaire is based on the TPB,[29] and a number of additional constructs from the Social Ecological Model[36] and the Habit Scale Index[37] were added. Figure 1 shows our extended TPB model. *Behaviour* (in this case self-reported compliance, a behaviour in previous internships) is influenced by *intention* (whether you intend or plan to comply with guidelines), which in turn is influenced by *attitudes* (beliefs of the importance of hand hygiene; outcome beliefs, that is, expected outcomes of hand hygiene); *social norms* (referent beliefs, that is, beliefs about how other people think about hand hygiene; descriptive norm, that is, the perceived behaviour of others) and *self efficacy* (the perception of whether students think they could perform hand hygiene). We added the following additional constructs:

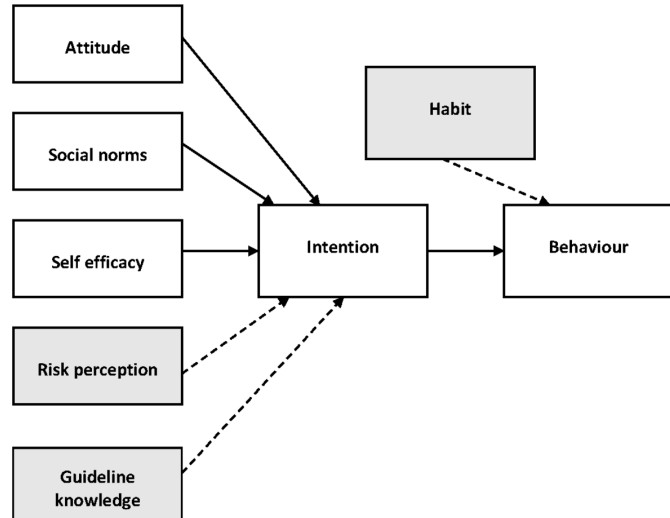

**Figure 1** Extended theory of planned behaviour model. Constructs in grey boxes have been added.

knowledge of the guidelines, risk perception (chance of infection occurring; severity of infection for self (ie, student) or patient) and habit (hand hygiene is something you do without thinking about it), measured with the self-report index of habit.[37] Since the internal consistency was at least adequate for each construct (Cronbach's α ≥0.70) we calculated average scores for use in further analysis (see table 1). All items, with the exception of the knowledge questions (measured by five true/false questions), were answered using 7-point Likert scales.

Self-reported compliance to hand hygiene guidelines, as outcome measure, was measured on a scale from 0 to 10 (never to always) for 13 potential hand hygiene situations (eg, before touching a patient, before wound care, after patient contact).[35] For each respondent, we calculated an average score for use in further analysis.

## STATISTICAL ANALYSIS
We performed hierarchical multivariate linear regression analysis to identify the effects of the behavioural factors (independent variables) on self-reported compliance (dependent variable). The constructs were added to the model in three steps:
1. Knowledge and risk perception,
2. All factors of step 1, with the addition of attitude, social norms and self-efficacy, and
3. All factors of step 2, with the addition of habit.
   Comparison of self-reported compliance scores between male and female students was done by means of t-test for between group comparisons.

## RESULTS
### Demographic data
In total, 313 (97%) students completed at least 75% of the questionnaire and were included in the analysis. The students had an average age of 25.3 years (SD 2.9), and 201 (64%) of the students were female, which is representative of the male-female ratio of medical students in the Netherlands.[38]

**Table 1** Constructs of the questionnaire on behavioural determinants of hand hygiene with example questions and internal consistency (Cronbach's α)

| Construct | # items | Mean (SD) | Cronbach's α | Example |
| --- | --- | --- | --- | --- |
| Knowledge | 5 | 4.3 (0.78) | – | five true/false questions about factual knowledge |
| Risk perception: | | | | |
| Chance | 3 | 5.6 (1.7) | 0.76 | How big is the chance that an infection will occur |
| Severity self | 1 | 5.6 (2.3) | – | How severe will the consequences of an infection be for myself |
| Severity patient | 1 | 7.2 (1.5) | – | How severe will the consequences of an infection be for my patient |
| Attitudes: | | | | |
| Beliefs about hand hygiene | 8 | 5.3 (0.81) | 0.76 | Hand hygiene is something I find important |
| Perceived outcomes | 5 | 5.0 (0.98) | 0.78 | If I follow that hand hygiene guidelines my patients will develop fewer infections |
| Social norms: | | | | |
| Referent beliefs | 4 | 4.7 (1.2) | 0.91 | My superior thinks that I should always follow the hand hygiene guidelines |
| Descriptive norm | 8 | 3.5 (0.81) | 0.73 | My colleagues always follow the hand hygiene guidelines |
| Self-efficacy | 11 | 4.8 (0.94) | 0.89 | I am certain that I will be able to follow the hand hygiene guidelines |
| Habit | 12 | 4.7 (1.1) | 0.95 | Following the hand hygiene guidelines is something I do without thinking about it |

**Table 2**  Behavioural correlates of hand hygiene compliance of medical students (n=313)

| | Model 1 | | Model 2 | | | Model 3 | | |
|---|---|---|---|---|---|---|---|---|
| | β | R² | β | R² | R²change | β | R² | R²change |
| | | 0.043 | | 0.270 | 0.189 | | 0.401 | 0.131 |
| Knowledge | 0.081 | | 0.044 | | | 0.063 | | |
| Risk perception: | | | | | | | | |
| Chance | 0.101 | | 0.049 | | | 0.010 | | |
| Severity self | 0.105 | | 0.041 | | | 0.002 | | |
| Severity patient | 0.102 | | 0.036 | | | 0.019 | | |
| Attitude: | | | | | | | | |
| Beliefs | | | 0.103 | | | −0.026 | | |
| Perceived outcomes | | | 0.231*** | | | 0.174** | | |
| Social norms: | | | | | | | | |
| Referent beliefs | | | 0.003 | | | −0.001 | | |
| Descriptive norm | | | 0.063 | | | 0.037 | | |
| Self-efficacy | | | 0.306*** | | | 0.138* | | |
| Habit | | | | | | 0.471*** | | |

Model 1: Knowledge + risk perception.
Model 2: Knowledge + risk perception + attitudes + social norms + self efficacy.
Model 3: Knowledge + risk perception + attitudes + social norms + self efficacy + habit.
*p<0.05; **p<0.01; ***p<0.001.

## SELF-REPORTED COMPLIANCE

The average self-reported compliance was 8.1 on a 10-point scale (SD .96). This measure was 8.2 (SD 0.92) for females and 7.8 (SD 0.99) for males. This difference was statistically significant (p<0.05). Self-reported compliance ranged from 4.3 (when resuming care after an interruption) to 9.8 (after direct contact with body fluids).

## BEHAVIOURAL CORRELATES

Table 2 shows the behavioural correlates associated with hand hygiene compliance in medical students. The regression coefficient β indicates the slope of the regression-line, and gives the average increase of compliance when the variable increases by 1. Knowledge of guidelines and risk perception explained 4.3% of the variance of self-reported compliance (adjusted $R^2$=0.043) (model 1). However, the contribution of these factors to self-reported hand hygiene compliance was not statistically significant.

The addition of attitude, social norms and self-efficacy (model 2), resulted in an explained variance of 27%: ($R^2$=0.270), with perceived outcomes (an element of attitude) (β=0.231, p<0.001) and self-efficacy (β=0.306, p<0.001) showing a statistically significant association with self-reported compliance.

In model 3, the addition of habit resulted in an explained variance of 40% (adjusted $R^2$=0.401), with habit showing a strong and statistically significant association with self-reported compliance, (β=0.471, p<0.001).

The associations of perceived outcomes and self-efficacy were somewhat weakened in this final model, but both remained statistically significant at the 5% level.

## DISCUSSION

The results of this study show that the hand hygiene behaviour of final year medical students, that is, the new generation of physicians, is most strongly influenced by habit, perceived outcomes of hand hygiene and whether students feel they have the ability to perform hand hygiene in practice. Our extended behavioural model, which included attitudes, social norms, self-efficacy, knowledge, risk perception and habit, was able to explain a substantial part of the variance in self-reported compliance (adjusted $R^2$=0.401).

One strength of our study is that we were able to include a full class of all medical students during 1 year, with a response of 97%. Students were approached to fill out the questionnaire after they had completed an 18-month rotation schedule of nine specialities in both teaching and non-teaching facilities in a mixed urban/rural area. This class therefore had recently experienced a large number of hospital settings and patient types. After graduation, the students may select any clinical or non-clinical speciality, and our study population therefore represents juniors that will continue their career within a broad spectrum of medical disciplines. A second strength of this study is that we used a hand hygiene questionnaire, based on combined insights from the Theory of Planned Behaviour and Social Ecological Models.

The use of self-reported compliance as the primary outcome measure and lack of observational data form one of this study's limitations, and we are therefore only able to base our model on self-reported behaviour. In the setting of an internship in Public Health, where this study was conducted, opportunities for hand hygiene are almost absent and directly observing hand hygiene compliance in the multitude of very diverse medical institutions during the preceding rotation schedule would not have yielded comparable data. This rotation schedule also resulted in the use of a cross-sectional design, which restricts conclusions on causality. A longitudinal study would resolve this restriction but would be arduous due to the rotations and most likely result in a high numbers lost to follow-up and a much lower response rate as a result. Therefore the use of self-reported data in a cross-sectional design was the best option in our case. Further, there is a possibility of social desirability bias since students completed the questionnaire in a class-room setting.[39]

A second limitation of our study arises from our inability to explore the influence of cultural factors in our analyses. International patient safety experts have addressed the need to tackle not only individual change but also organisational change in order to improve patient safety culture.[1] A poor safety culture has been found to be associated with adverse events and a substantial improvement requires a culture of safety within the organisation.[40] We would therefore have liked to include the construct culture in our model, but due to the large number of wards within different hospitals that students worked on (and therefore large variation in culture they might have experienced) we were unable to investigate its effects in this study. Culture could prove a valuable addition to the model and explain an additional part of the behaviour of medical students. The influence of culture should be further investigated in a different study design focussing on the observed compliance of interns of specific units, hospitals and/or specialities.

As a third limitation it must be mentioned that, although we used an extended version of the Theory of Planned Behaviour, potential correlates of hand hygiene compliance could have been omitted. Other studies have used more comprehensive models, such as the Theory Domains Framework[41] and the Health Action Process Approach[42] to explain hand hygiene compliance of physicians and nurses and their application in medical students could be considered. Although the addition of the Habit Scale Index did add an extra 13% explained variance, indicating hand hygiene is a strongly habitual behaviour, and interventions to improve it should not only focus on volitional construct

Only a few studies on the observed hand hygiene compliance of medical students have been conducted so far, mostly looking at other factors than student behaviour (eg, facilities), limiting a straightforward comparison of the results presented here.[30 31] We found that in addition to external factors such as access to facilities and compliance of superiors, student-related behavioural factors

make their contribution as well. Previous research has shown that already at an undergraduate level difference between perception and knowledge towards hand hygiene can exist (with nurse more scoring more positively),[16] which we found also. A study from the UK found that the observed hand hygiene guideline compliance of medical students in an examinational setting was extremely low, even in the presence of 'Wash Your Hands' signs.[43] A hand hygiene intervention after the SARS (severe acute respiratory syndrome) outbreak in Asia had good results, and was found to be related to a higher level of perceived risk; risk perception was not found to be significantly associated with hand hygiene compliance in this study, although this difference could be a result of the extreme situational circumstances during the SARS outbreak.[44]

Previously positive attitudes towards hand hygiene, and in particular positive beliefs about the outcome of performing hand hygiene have been found to be significantly associated with hand hygiene compliance of nurses and physicians,[23 45 46] similar to the results we found here for medical students. The influence of habit[46] on hand hygiene behaviour had been understudied so far, and we found a strong association between habit and the self-reported hand hygiene behaviour of medical students. Since habit seems to play an important role in influencing hand hygiene behaviour, and the foundation for these professional habits is laid down during medical training, it is important for the working environment of junior doctors to further stimulate strengthening these habits. Habit is a complex construct, referred to as a 'habitual mind-set' in which people focus less on new information, but rather fall back on previously formed automatic cue-responses, thereby maintaining that behaviour.[46] Actions have to be repeated often enough in a stable context in order to shape a habitual mindset. The behaviour then becomes automatic and could even overrule intentions.

Much of the behaviour of medical students is based on the behaviour of the role models (often residents) they encounter during their clinical phase, and not on what they have learnt during their preclinical phase.[18 47 48] Once they reach their internship-phase, medical students are confronted with and adapt themselves to a culture of non-adherence. This effect is also present among residents, as with a senior member of the team performing hand hygiene, the hand hygiene compliance of residents increases significantly, but overall compliance of residents is as low as their qualified colleagues (<40%).[47] It is therefore essential to break the vicious circle and one way to do this is by preparing medical students for the incongruences they will encounter in clinical practice and increase their coping skills. Based on habit theory, in the case of hand hygiene, it could be essential to shape the correct mindset in the correct context (eg, on the workfloor, as opposed to the class room) in order for strong habits to be formed.[46]

It is increasingly recognised that patient safety should be improved through education.[19 34] Medical students

indicate themselves that more education on patient safety and especially hand hygiene is necessary.[49] Residents state that medical mistakes could be prevented with more education on the matter.[50] Our results show that traditional educational methods focussing on knowledge improvement are not the way to go in order to stimulate better patient safety behaviours, such as hand hygiene compliance. Similarly to physicians and nurses it is essential to target medical students with interventions tailored to the major modifiable determinants of non-compliance. Targeting medical students' behaviour should focus on the empowerment of these juniors, rather than increasing their factual knowledge of procedures. Insights from the behavioural sciences may be useful to increase the self-efficacy of this important target group. Interventions using the concept of action planning or implementation intentions have been successful in several settings,[51 52] including hospital care[53] and seem to be promising in this context.

Adequate hand hygiene can lead to a reduced rate of HAI and a drop in adverse events, morbidity and mortality. Application of behavioural insights can lead to patient safety improvements throughout healthcare, and ultimately to safer hospitals. Breaking through the culture of non-adherence is the first step to achieving this goal.

In conclusion, targeting medical students' behaviour should focus on the empowerment of these juniors and provide them with evidence on the health benefits of prevention, rather than increasing their factual knowledge of procedures. Clinical teaching environments could help them form good patient safety habits during this vital phase of their career.

**Acknowledgements** The authors wish to thank the medical students for their participation in this study.

**Contributors** The study was conceived and designed by VE and EB. Data collection was performed by VE and RE and analysed by VE. VE, SO, ER, RE, MV, AB and EB were involved in the interpretation of the data. VE wrote the first draft of the manuscript. VE, SO, ER, RE, MV, AB and EB critically revised the manuscript. VE, SO, ER, RE, MV, AB and EB read and approved the final manuscript.

**Funding** This work was supported by the Netherlands Organisation for Health Research and Development (*ZonMW*); grant number 50-50115-96-632.

**Competing interests** None declared.

**Patient consent for publication** Not required.

**Provenance and peer review** Not commissioned; externally peer reviewed.

**Data availability statement** All data relevant to the study are included in the article or uploaded as supplementary information. No additional data are available.

**ORCID iDs**
Suzie Otto http://orcid.org/0000-0002-9171-2251
Alex Burdorf http://orcid.org/0000-0003-3129-2862

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
