## [Reviewer comments · BMJ Open]

ARTICLE DETAILS

TITLE (PROVISIONAL)	Assessment of correlates of hand hygiene compliance among final year medical students: a cross-sectional study in the Netherlands
AUTHORS	Erasmus, Vicki; Otto, Suzie; De Roos, E; van Eijdsen, Rianne; Vos, Margreet; Burdorf, Alex; VAN BEECK, Ed

VERSION 1 - REVIEW

REVIEWER	Thea van de Mortel Griffith University I have published on hand hygiene of medical students.
REVIEW RETURNED	12-Mar-2019

GENERAL COMMENTS	An interesting study which adds to the knowledge of factors influencing hand hygiene of medical students. Suggest the following revisions to improve clarity and depth: • Edit for English expression and punctuation. I have scanned an annotated copy with some suggested edits.• Introduction: incorrect statement that studies into medical students HH lacked a theoretical framework. For example, there are a number of publications by van de Mortel et al that used a survey tool to examine med students HH that was underpinned by Social Cognitive Theory. The HHQ covered a range of factors including habit and looked at students' beliefs, and practices and how they were educated on HH. Suggest you make your lit review more comprehensive• Methods: were students recruited by email? In person? By the teacher? By an RA? Were they given an information sheet? How was content validity and face validity determined? Was the survey anonymous? online? paper? how do you do verbal consent to a survey? suggests that students did not really have an option to decline to participate (ie power relationships between the researcher and participants). Were they watched while they completed the survey (I am guessing it was paper based and handed out in person?)• Cronbach's alpha of 0.7 is considered adequate not good• Item to total correlations are considered more appropriate for scales with fewer than 10 items• Discussion: Consider socially desirable responding as a limitation for your survey particularly if the researcher was watching students complete the survey. consider the issues around ability to self-assess (also shown to be less than stellar).
--

	 • Incorrect to state that habit has not previously been included in studies of HH behaviour, again see van de Mortel. • Could see Ng et al re influence of culture on HH • Discussion would be strengthened by greater reference to literature to put findings into context of what is already known about medical students HH and influencing factors The reviewer provided a marked copy with additional comments. Please contact the publisher for full details.
--	--

REVIEWER	Katie Page University of Technology Sydney, Australia
REVIEW RETURNED	30-Sep-2019

GENERAL COMMENTS	Dear Authors, Thank you for this interesting research which has clear interest and merit. Targeting medical students is extremely important in breaking the cycle of non-compliance to hand hygiene guidelines. For the introduction I have a few suggestions: I think it is important to say that understanding how the behaviours are formed are very important for designing interventions to alter behaviour in the early stages of a career. Changing behaviour once habits have been formed is much harder. I think this provides an additional important rationale for concentrating on medical students. It might also be worthwhile stating why medical students as opposed to other types of students are so important. You could quote work saying that the medical practitioners often have lower compliance to hand hygiene guidelines than other types of health professionals (i.e. nursing staff). P.5 line 56-58 – could you give a little bit more detail about the Basic Hygiene Unit – and what they are taught re hand hygiene. For example – what length of time would they spend learning about it? Are there any practical learning or classes on rotation etc? Is the teaching all theoretical? Are there cases studies etc? Other general comments/suggestions: What is the rationale for including other constructs like habit, knowledge and risk perception? It seems more obvious with the latter two but a sentence or two on this would be helpful to the readers I think. The limitations of the study are acknowledged quite well. Clearly the large one is the establishment of causality with a cross-sectional study and a self-reported outcome. The need for good longitudinal data and observational dependent variables is pressing. It would be worthwhile to be able to track these students as they enter their specialities to see how their beliefs, attitudes and behaviour change over time and with changing hospital environments. Could you make a comment on this?
--

	Given that habit seems to be an important determinant could you say something more on this. These students are all very new the hospital practice so how are these habits formed? Are they personal habits that flow over into their work environment? Do they change over time? What suggestions do you have to altering these habits? Is it possible that habit is just like a decision heuristic which is a combination off some of the other constructs in the TPB? I suggest a short theoretical consideration of the distinctness of these constructs is warranted perhaps with reference to Verplanken B, Aarts H. Habit, Attitude and Planned Behaviour: Is Habit an Empty Construct or an Interesting Case of Goal-Directed Automaticity? . For the reference to the Self Report Index of Habit - it seems wrong. It should be this one: Verplanken, B. and Orbell, S., 2003. Reflections on past behavior: a self-report index of habit strength 1. Journal of applied social psychology, 33(6), pp.1313-1330. Could you also state some of the evidence-based ways (informed from your findings and others) in which medical training could alter to increase student compliance with the guidelines? You mention action-planning but it is unclear what you mean by this. Could you give an example or provide more detail? Thank you for your research in this important area and all the best with its continuation. Kind regards, Katie Page
--	--

VERSION 1 – AUTHOR RESPONSE

Reviewer: 1

An interesting study which adds to the knowledge of factors influencing hand hygiene of medical students. Suggest the following revisions to improve clarity and depth:

- Edit for English expression and punctuation. I have scanned an annotated copy with some suggested edits.

Many thanks for the suggested edits. They have been included, in addition to a full screening of the paper.

- Introduction: incorrect statement that studies into medical students HH lacked a theoretical framework. For example, there are a number of publications by van de Mortel et al that used a survey tool to examine med students HH that was underpinned by Social Cognitive Theory. The HHQ covered a range of factors including habit and looked at students' beliefs, and practices and how they were educated on HH. Suggest you make your lit review more comprehensive

We agree this was misstated. We have adjusted the text throughout the paper (e.g. see page 5, main document-marked copy) and added more literature (refs 16, 32, 33).

- Methods: were students recruited by email? In person? By the teacher? By an RA? Were they given an information sheet? How was content validity and face validity determined? Was the survey anonymous? online? paper? how do you do verbal consent to a survey? suggests that students did not really have an option to decline to participate (ie power relationships between the researcher and participants). Were they watched while they completed the survey (I am guessing it was paper based and handed out in person?)

More details about the data collection methods have been added (see pages 5-7, main document-marked copy). Although we did recruit students in a classroom setting, we did take precautions to avoid influencing students into participation and or in the answers they gave.

- Cronbach's alpha of 0.7 is considered adequate not good

We have adjusted the sentence accordingly to reflect the Cronbach alpha's of 0.73-0.95 (page 7, main document-marked copy).

- Item to total correlations are considered more appropriate for scales with fewer than 10 items

We thank the reviewer for this suggestion. We examined the item to total correlations, and these were well acceptable for all but one item (construct chance, item: how big is the chance that an infection will occur in your hospital). The item-total correlation for this item was 0.35 was low and excluding it would increase the overall Cronbach's alpha. However, we prefer to keep it in the construct as it does cover a part of risk perception otherwise excluded. Other item-total correlation were well above 0.7.

- Discussion: Consider socially desirable responding as a limitation for your survey particularly if the researcher was watching students complete the survey. consider the issues around ability to self-assess (also shown to be less than stellar).

See text added to discussion on page 10, main document-marked copy. The limitations of self-reported behavior are also addressed in the discussion.

- Incorrect to state that habit has not previously been included in studies of HH behaviour, again see van de Mortel.

We have adjusted the discussion accordingly (page 12, main document-marked copy).

- Could see Ng et al re influence of culture on HH

In this paper we refer to safety culture within the hospital. We have clarified this in the manuscript.

- Discussion would be strengthened by greater reference to literature to put findings into context of what is already known about medical students HH and influencing factors

We have added some sentences to the discussion to include more context (see page 11, main document-marked copy).

Reviewer: 2

Thank you for this interesting research which has clear interest and merit. Targeting medical students is extremely important in breaking the cycle of non-compliance to hand hygiene guidelines

For the introduction I have a few suggestions:

- I think it is important to say that understanding how the behaviours are formed are very important for designing interventions to alter behaviour in the early stages of a career. Changing behaviour once habits have been formed is much harder. I think this provides an additional important rationale for concentrating on medical students.

It might also be worthwhile stating why medical students as opposed to other types of students are so important. You could quote work saying that the medical practitioners often have lower compliance to hand hygiene guidelines than other types of health professionals (i.e. nursing staff).

This has been added to the text (see pages 4 and 5, main document-marked copy).

- P.5 line 56-58 – could you give a little bit more detail about the Basic Hygiene Unit – and what they are taught re hand hygiene. For example – what length of time would they spend learning about it? Are there any practical learning or classes on rotation etc? Is the teaching all theoretical? Are there cases studies etc.?

We have added a few words on the basic hygiene training, which is unfortunately quite limited.

Other general comments/suggestions:

- What is the rationale for including other constructs like habit, knowledge and risk perception? It seems more obvious with the latter two but a sentence or two on this would be helpful to the readers I think.

Indeed this addition is valuable for readers. We have added some explanation. In a previous qualitative study among physicians and nurses, these aspects were identified as important in influencing hand hygiene behavior. Furthermore, face and content validity were assessed by a team of experts in the field of behavioral sciences and infection control.

- The limitations of the study are acknowledged quite well. Clearly the large one is the establishment of causality with a cross-sectional study and a self-reported outcome. The need for good longitudinal data and observational dependent variables is pressing. It would be worthwhile to be able to track these students as they enter their specialties to see how their beliefs, attitudes and behavior change over time and with changing hospital environments. Could you make a comment on this?

Yes, we would love to track students over time. So far, we have been unable to do so, as after training most students leave the hospital and even those that stay, complete their specialists training in various hospitals. Finding people at a certain point in time is quite tricky. This would be a valuable addition in the future.

- Given that habit seems to be an important determinant could you say something more on this. These students are all very new the hospital practice so how are these habits formed? Are they personal habits that flow over into their work environment? Do they change over time? What suggestions do you have to altering these habits?

Hand hygiene habits can be personal (people who always like to wash/clean their hands) or professional (when at work people wash/clean their hands, at home they do not). There is literature suggesting that personal habits are shaped during childhood (e.g. at home, school) but professional habits can be shaped later on in life. It is important though that junior doctors then work in an environment enabling good hand hygiene behavior, through positive reinforcement and good safety culture and good facilities. Once habitual mind-sets are shaped in the correct context, they can be better sustained in shifting departments.

- Is it possible that habit is just like a decision heuristic which is a combination off some of the other constructs in the TPB? I suggest a short theoretical consideration of the distinctness of these constructs is warranted perhaps with reference to Verplanken B, Aarts H. Habit, Attitude and Planned Behaviour: Is Habit an Empty Construct or an Interesting Case of Goal-Directed Automaticity? .

We thank the reviewer for this very interesting comment, and have added this insight to the discussion (pages 12, 13, main document-marked copy). We feel it brings more depth to the discussion and hope it will provide useful for future readers.

- For the reference to the Self Report Index of Habit - it seems wrong. It should be this one:

Verplanken, B. and Orbell, S., 2003. Reflections on past behavior: a self-report index of habit strength 1. *Journal of applied social psychology*, 33(6), pp.1313-1330.

Indeed, somehow the wrong reference was included. Many thanks for identifying this; we have corrected it (ref 37).

- Could you also state some of the evidence-based ways (informed from your findings and others) in which medical training could alter to increase student compliance with the guidelines? You mention action-planning but it is unclear what you mean by this. Could you give an example or provide more detail?

Action-planning is a method in which one can bridge the intention-behavior gap by formulating concrete actions, linked to time and context, thereby creating cues as it were. E.g. instead of stating you would like to exercise more, you formulate it as (linked to some other behavior you already have): after I watch the 8 o'clock news I will put on my running shoes and go for a 10 minute run around the block. (Sniehotta FF, Scholz U, Schwarzer R. Bridging the intention-behavior gap: planning, self-efficacy and action control in the adoption and maintenance of physical exercise. *Psychol Health*. 2005;20:143–60.)

VERSION 2 – REVIEW

REVIEWER	Thea van de Mortel Griffith University, Australia
REVIEW RETURNED	12-Nov-2019

GENERAL COMMENTS	thank you for addressing the comments. Two small changes: Pg 29 line 17 van de Mortel not van der Mortel Provide a ref for social desirability bias pg 10 line 35
---

REVIEWER	Dr Katie Page CHERE (Centre for Health Economics Research and Evaluation)
REVIEW RETURNED	02-Dec-2019

GENERAL COMMENTS	The authors have adequately addressed the issues that I raised in the first review. A couple of minor points remain: There is a sentence that has been added (p.7 line 5-7) about experts in the field of behavioural science and infection control examined all items etc... It would be good to state how many experts. To the addition of social desirability bias (p.10 line 32-24) is the concern with self selection of students into the study which likely overestimate any effects on self-reported behaviour that have been found. The keener and more motivated students opt it. A statement on this would be prudent. Thank you and best of luck with your continuing research program. Katie
--

VERSION 2 – AUTHOR RESPONSE

Reviewer: 1

thank you for addressing the comments. Two small changes:

Pg 29 line 17 van de Mortel not van der Mortel

Many thanks for identifying this; we have corrected it.

Provide a ref for social desirability bias pg 10 line 35

We added the paper 'van de Mortel, T. 2008. Faking it: social desirability response bias in self-report research. Australian Journal of Advanced Nursing, 25(4):40-48 ' as reference.

Reviewer: 2

The authors have adequately addressed the issues that I raised in the first review. A couple of minor points remain:

There is a sentence that has been added (p.7 line 5-7) about experts in the field of behavioural science and infection control examined all items etc... It would be good to state how many experts.

3 experts in the field of behavioural science, and 3 in the field of infection control; this information has been added in the text.

To the addition of social desirability bias (p.10 line 32-24) is the concern with self selection of students into the study which likely overestimate any effects on self-reported behaviour that have been found. The keener and more motivated students opt it. A statement on this would be prudent.

Since we achieved a response rate of 97% we do not feel there would be much effect of self-selection on the outcomes of the study. Although of course in general keener student choose to participate in studies voluntarily and with lower response rates this could be a serious form of bias.